# Application of Multi-Layered Temperature-Responsive Polymer Brushes Coating on Titanium Surface to Inhibit Biofilm Associated Infection in Orthopedic Surgery

**DOI:** 10.3390/polym15010163

**Published:** 2022-12-29

**Authors:** Sookyung Choi, Hyeonjoon Lee, Ran Hong, Byungwook Jo, Suenghwan Jo

**Affiliations:** 1School of Medicine, Chosun University Medical School, Gwangju 61452, Republic of Korea; 2Department of Orthopedic Surgery, Chosun University Hospital, Gwangju 61453, Republic of Korea; 3Department of Pathology, Chosun University Hospital, Gwangju 61453, Republic of Korea; 4School of Engineering, Chosun University, Gwangju 61452, Republic of Korea

**Keywords:** antibiotics coating, bacterial biofilm, controlled release, lower critical solution temperature, periprosthetic joint infection, rodent model, smart coating, thermo-responsive

## Abstract

Infection associated with biomedical implants remains the main cause of failure, leading to reoperation after orthopedic surgery. Orthopedic infections are characterized by microbial biofilm formation on the implant surface, which makes it challenging to diagnose and treat. One potential method to prevent and treat such complications is to deliver a sufficient dose of antibiotics at the onset of infection. This strategy can be realized by coating the implant with thermoregulatory polymers and triggering the release of antibiotics during the acute phase of infection. We developed a multi-layered temperature-responsive polymer brush (MLTRPB) coating that can release antibiotics once the temperature reaches a lower critical solution temperature (LCST). The coating system was developed using copolymers composed of diethylene glycol methyl ether methacrylate and 2-hydroxyethyl methacrylate by alternatively fabricating monomers layer by layer on the titanium surface. LCST was set to the temperature of 38–40 °C, a local temperature that can be reached during infection. The antibiotic elution characteristics were investigated, and the antimicrobial efficacy was tested against *S. aureus species* (Xen29 ATCC 29 213) using one to four layers of MLTRPB. Both in vitro and in vivo assessments demonstrated preventive effects when more than four layers of the coating were applied, ensuring promising antibacterial effects of the MLTRPB coating.

## 1. Introduction

Orthopedic surgeries commonly require the insertion of metallic implants within the bone and under the soft tissue, which may result in alteration of the host immune response and exposure to pathogens [1,2]. When the host defense mechanism is weakened or an incidental increase in bacterial bioburden occurs, planktonic bacteria may adhere to the implant, leading to periprosthetic infection (PJI) [3,4]. The adherent bacteria aggregate and develop an extracellular polymeric matrix (EPM) forming a bacterial biofilm. The intractable nature of PJI is largely attributed to the inability to clean the bacterial biofilm adherent to the implant [5,6,7]. As mature biofilms are resistant to systemic antibiotics and the host immune system, subsequent surgery is often necessary, which involves removal of the implant and reinsertion once the infection has been eradicated. Such complications can be observed after any implant insertion surgery, such as arthroplasty or fracture fixation, resulting in significant costs and morbidity for the patient [8,9].

Numerous efforts were made to prevent and eradicate bacterial biofilms. A common strategy is the chemical or structural modification of the implant surface with or without the use of pharmacologically active substances [10,11,12,13,14,15]. A clinical application has been made with the use of iodine, silver nanoparticles, Poly(D, L-lactide) (PLLA), and hydrogel coating composed of covalently linked hyaluronan [16,17]. These coating methods have been shown to provide antibacterial efficacy, but cytotoxicity has been suggested to be a potential complication. Recently, coating implants with antibiotics has been widely employed, as the antibiotic coating can theoretically prevent planktonic bacteria from adhering to the implant, thereby inhibiting the development of bacterial biofilms. However, such attempts had a drawback in being used clinically as most of the antibiotics were released immediately following implant insertion and thus had limited effect on the prevention of subacute or chronic infection [16,17]. More recently, a “smart” coating that passively elutes the antibiotic with an active-release mechanism was introduced using branched poly(ethylene glycol)-poly(propylene sulfide) (PEG-PPS) polymer, which promotes antibiotic release by the reactive oxygen cascade initiated by bacteria [16]. While the PEG-PPS coating showed promising in vitro and in vivo antibacterial efficacy, its clinical application was limited as it was designed to be short-acting and biodegradable and therefore effective only to prevent infection during or immediately following the surgery. 

An optimal method to prevent bacterial biofilms would be to allow antibiotic release only when an infection is anticipated. One of the early reactions to infection is the development of heat from the inflammatory process, which can occur both systemically and locally at the infected site. Therefore, it can be hypothesized that if the antibiotics are coated on the implant and released when the fever has occurred, this may provide a potential solution to prevent bacterial biofilm formation. Thermo-regulatory polymers can respond to temperature change and have been utilized for biomedical applications such as in the delivery of therapeutic molecules or tissue engineering [18]. As lower critical solution temperature (LCST) could be controlled by adjusting the composition ratio of HEMA and OEGMA, developing a coating strategy through a combination of the two polymers may fulfill our purpose [19]. A similar methodology has been attempted previously using thermo-responsive poly(di(ethylene glycol) methyl ether methacrylate) (PDEGMA) brushes on titanium implants synthesized via a surface-initiated atom transfer radical polymerization with activators regenerated through the electron transfer technique (ARGET ATRP) with promising results [20]. However, the dose of antibiotics loaded on the implant was limited because of the amount and short length of the coated polymer brush. 

Herein, we introduce a novel coating method that utilizes a thermo-responsive polymer brush to prevent bacterial adhesion and biofilm formation on titanium surfaces. The surface-grafted polymer films in aqueous media can be amplified by covalently layering the thermosensitive brushes. Therefore, we are using a strategy of fabricating polymer brushes in multilayer architectures featuring diverse hydrophilicities and compositions to optimize antibiotic delivery to implant surfaces. Here, we report the efficacy of our coating method using both in vitro and in vivo results.

## 2. Materials and Methods

The multi-layered temperature-responsive polymer brushes (MLTRPB) on titanium (Ti) implants were fabricated using poly(ethylene glycol). Methyl ether methacrylate (OEGMA, Aldrich, Mn; 300 g/mol), di(ethylene glycol) methyl ether methacrylate (DEGMA, Sigma-Aldrich, St. Louis, MO, USA, Mw; 188.22 g/mol), and 2-hydroxyethyl methacrylate (HEMA, Sigma-Aldrich, St. Louis, MO, USA) by atom transfer radical polymerization (ATRP). Levofloxacin (LVF, USP, Israel) and gentamycin (GM, 40 mg/mL, Shinpoong Pharmaceutical Co., Ltd., Seoul, Korea) were used to test the efficacy of MLTRPB. Both agents have an antibacterial effect against different types of Gram-positive and Gram-negative bacteria, especially against the S aureus strain, the most common pathogen after orthopedic implantation [21,22,23]. 

### 2.1. Fabrication of MLTRPB on Titanium Surface

To prepare the MLTRPB on titanium alloy (Ti-6Al-4V) implants, the surfaces of Ti implants were treated by surface-initiated atom transfer radical polymerization with activators regenerated through the electron transfer technique (SI-ARGET ATRP), as described previously [20]. MLTRPBs were fabricated on two different Ti implants: (1) disk-type titanium plates (10 mm diameter and 1 mm thickness, 1.57314 cm^3^ area) were used to measure the amount and thickness of the coated MLTRPB and to test the elution properties, and (2) cannulated screws (10 mm length, 25 mm head width, average weight 908.5 μg) were used to test the in vitro and in vivo antibacterial efficacy of MLTRPB.

The MLTRPB was prepared on multi-layered films comprising linear polymer “brush” layers, based on poly(DEGMA_90_-co-OEGMA_10_) and poly(HEMA_20_-co-OEGMA_80_) layers (Figure 1). The first layer was fabricated with DEGMA_90_-co-OEGMA_10_ in 40 mL of Me-OH/Milli-Q water, and then the same method was used to fabricate the second layer composed of HEMA_20_-co-OEGMA_80_. This process was repeated until the 4th layer. The 3rd and 4th polymerization were carried out with 30 mL of Me-OH/Milli-Q water. In order to control the LCST of MLTRPB to 38–40 °C the molar ratio of the monomer in polymerization was set to 0.9/0.1 for DEGMA/OEGMA and 0.2/0.8 of HEMA/OEGMA, so DEGMA rich layers are 1st and 3rd layers, and HEMA rich layers are 2nd and 4th layers [19,24]. After polymerization of MLTRPB on Ti implants (MLTRPB-TI), the substrates were washed with water and Et-OH and dried thoroughly under vacuum for 24 h. The thickness of the MLTRPB was studied using ellipsometry, and field-emission scanning electron microscopy (FE-SEM, S-4800, Hitachi) was used when the thickness exceeded 1 μm. The amount of MLTRPB on the Ti implants was estimated by measuring the mean weights of 20 specimens per group of different layers and calculating the difference among the groups using an electronic scale that can be measured in detail up to 10^−4^ g.

The fabrication of the MLTRPB was characterized by HP-XPS (High-Performance X-ray Photoelectron Spectroscopy, ThermoFisher Scientific, Waltham, MA, USA, K-ALPHA+) and FT-IR (FT-infrared spectroscopy, ATR mode, ThermoFisher Scientific, iS10). The hydrophilicity of the MLTRPB was determined using an automatic contact angle analyzer (Phoenix 300 Touch, SEO, Kromtek, Shah Alam, Malaysia) measured at room temperature using a standard test method for corona-treated polymer films (ASTM D5946). 

### 2.2. In Vitro Antibiotic Elution Kinetics

To load the antibacterial agent on the MLTRPB-Ti implants, the fabricated implant was first dried thoroughly and immersed in 4 mL of 10 mg/mL LVF in Milli-Q water or 4 mL of 20 mg/mL GM diluted in D_2_O at room temperature. The loading process was performed under a vacuum for 1 h, followed by drying under a nitrogen stream, and then kept under a vacuum for over 12 h. As the hydrogen bonding between antibiotics and polymer brush influences the antibiotic loading dose, this process allows complete dehydration of MLTRPB which provides a uniform loading atmosphere for each sample. 

The LCST was confirmed by testing antibiotic release from MLTRPB-Ti implants at different temperatures (25, 40 °C). MLTRPB-Ti with LVF was loaded into the 5 mL of pH 7.0 PBS buffer, to imitate the inflammatory atmosphere. 0.5 mL of the suspension was retrieved at 0.5, 1, 2, 4, and 6 h and the concentration of LVF release (μg mL^−1^) was measured using fluorescence spectroscopy (Cary Eclipse spectrometer, Agilent, Santa Clara, CA, USA) at excitation and emission wavelengths of 292 nm and 540 nm and calculated using an established calibration curve [20]. The GM released from MLTRPB-Ti into PBS buffer-d_2_ (PBS was dissolved in D_2_O) at pH 7.0 was estimated at 0.5, 1, 2, 4, and 6 h using FT-NMR (JEOL 300 MHz, Tokyo, Japan) by measuring the peak area (1.84 ppm), which belongs to GM, at each time points. 

### 2.3. In Vitro Antibacterial Tests

*S. aureus* (ATCC 29213, Xen29) was used to determine the antibacterial properties of MLTRPB-Ti-LVF and MLTRPB-Ti-GM. Moreover, MLTRPB-Ti without antibiotics was used as a control. All substrates were sterilized by washing with 70% ethanol and drying under laminar flow. *S. aureus* was prepared by incubating overnight in tryptic soy broth (TSB) at 37 °C in a shaking incubator at 200 rpm. The dose of bacteria was confirmed using optical density (OD) and by measuring CFUs after serial dilutions of bacterial suspensions in TSB (10^2^ to 10^7^ fold dilutions) in duplicate.

MBC (minimum bactericidal concentration) of LVF and GM against *S aureus* was tested according to the established method by incubating LVF or GM (1, 2, 4, 10 μg) in 1 mL of TSB with *S. aureus* (2.0 × 10^6^ CFUmL^−1^) at 37 °C for 24 h [20,25]. To determine the antibacterial properties according to temperature, 1 mL of adjusted suspension (2.0 × 10^6^ CFU mL ^−1^) was co-cultured with MLTRPB-Ti with/without LVF and GM in six-well plates. After 0.5, 1, 2, 4, and 12 h of treatment at 25, 36.5, and 40 °C, 50 μL of the suspension was inoculated on the TSB agar and incubated for 24 h. Antibacterial properties were measured by counting CFUs and by measuring the luminescence intensity of bacterial luminescence (*S. aureus* Xen29) using the IVIS spectrum (In Vivo Imaging System, Perkin Elmer, Waltham, MA, USA) [26]. Biofilm formation on the titanium surfaces with and without antibiotic coating was confirmed by FE-SEM. 

### 2.4. Test for Cytotoxicity

Cytotoxicity of the MLTRPB was tested using a cell viability assay [27,28]. Briefly, mouse osteoblast cells (MC3T3-E1) were cultured in alpha-MEM, and 25 mM D-mannitol was added to minimize osmotic pressure on the cells. The assay was based on the reduction in MTT(3-(4,5-dimethylthiazol-2-yl)-2,5-diphenyltetrazolium bromide) by mitochondrial dehydrogenase in viable cells to produce a purple formazan product, which indicates the level of cell respiration. Using 96-well plates, cells were seeded at 5 × 104 cells per well with 200 μL of alpha-MEM. MLTRPB was diluted with DMSO and 10 μL of suspensions of various concentrations were added to the cells and cultured at 37 °C for 24 h. After removing the medium, the cells were incubated in phosphate-buffered saline containing 30 μL of MTT for 3 h. The formazan product was dissolved in 50 μL of dimethyl sulfoxide (Calbiochem, San Diego, CA, USA) and the OD was measured at 570 nm using a colorimetric microplate reader (BioTek, Winooski, VT, USA). 

### 2.5. In Vivo Validation

A rodent model was developed to test the in vivo antibacterial effects of the proposed coating. In consistence with the in vitro part of the study, *S Aureus* strain was selected for the test, as it is the most common pathogen following orthopedic surgery and has been used widely in preclinical animal models [29]. 

#### 2.5.1. In Vivo Assessment of MLTRPB Coating

All animal studies were performed after the approval of the experimental protocol from the Institutional Animal Care and Use Committee (IACUC). Six-week-old Sprague Dawley (SD) rats were used for the in vivo efficacy experiments. The mean weight of the rats at the time of the surgery was 221 ± 19 g. Rats were anesthetized with 3% isoflurane in 1 L of 0_2_/min. Using a sterile technique, a 3 cm skin incision was made over the knee and the knee joint was exposed. The femoral cartilage was partially removed using a surgical 1 mm drill tip, and an 18-gauge needle was inserted through this hole to identify the intramedullary canal. For the inserted implant, a 1.2 mm diameter cannulated screw was prepared (Trident medical, 1.2 mm in diameter and 10 mm in length). This screw was selected because of its small diameter and cannulated shape; however, additional roughening was performed on the screw surface using a drill tip to promote osteointegration. The MLTRPB coating on the screw was performed as described in the previous section, and the GM was impregnated into the MLTRPB. The screw was inserted until the screw head was in level with the articular surface of the distal femur (Figure 1). Following screw insertion, 10^6^ S aureus were injected into 30 μL solution on the screw cannula. The extensor mechanism and skin were closed layer by layer, and following the skin closure, thermal light was applied for 30 min so that the topical skin temperature of the rat’s knee reached 40–45 °C. All animals were euthanized on the 14th day following the surgery. Forty animals underwent surgery, which were divided into four groups of 10: Control with insertion of the non-fabricated screw, group I with MLTRPB in one layer, group II with MLTRPB in three layers, and group IV in four layers. 

#### 2.5.2. Measuring Bacterial Bioburden

The number of bacteria adherent to the screw was estimated by counting the number of colony-forming units after sonication. The femoral screw was aseptically retrieved and rinsed with PBS to remove planktonic bacteria. This was then placed in a 15 mL centrifuge tube containing 4 mL of PBS solution, which was vortexed for 30 s and sonicated for 15 min at 24 °C (52 kHz, 137 watts; BioSonic UC125; Coltene, Altstatten, Switzerland). The resulting solution was serially diluted and plated overnight on TSB agar in triplicate. The number of CFUs was quantified by two investigators who were blinded to each other after 24 h of incubation. 

In addition, soft tissue adjacent to the implant was harvested and fixed in 4% formaldehyde for 48 h. The sections were then stained with hematoxylin and eosin (H&E). The purpose of this process was to count the number of polymorphonuclear leukocytes (PMNL) in the soft tissue to validate whether eluted antibiotics reduced the bacterial bioburden in the adjacent pericapsular soft tissue. The number of PMNL was counted in the high-powered field of view using an Olympus BX 51 microscope by a pathologist.

## 3. Results and Discussion

### 3.1. Characterization of MLTRPB on Ti Implants

XPS and FT-IR were used to confirm the chemical structure of the polymer brush polymerized on the Ti implants. Different characteristic peaks of rich DEGMA in 1,3-MLTRPB and HEMA contained 2,4-MLTRPB were identified. In Figure 2A, three-layered MLTRPB(1-MLTRPB) on the implant, which is DEGMA rich layer, showed three peaks: C-C at 285.5 eV, C-O at 286.5 eV, and O-C=O at 288.7 eV in C1s region of the high-resolution XPS scan. The HEMA containing four-layered MLTRPB(-MLTRPB) on the implant has a stronger C-C at 285.5 eV compared to the C-C intensity of 3-layered MLTRPB(3-MLTRPB) on the implant. LVF-loaded 3-layered MLTRPB(-MLTRPB) has a C-F peak at 686 eV (Figure 2C), and GM-loaded 4-layered MLTRPB(-MLTRPB) has an N-H peak at 401.5 eV (Figure 2D).

As shown in the FT-IR spectra (Figure 3), the O-H bending vibration at 1170 cm^−^^1^ belongs to the hydroxy of HEMA-contained MLTRPB-layered Ti implant surface. This is consistent with the XPS data, and it was confirmed that the expected polymer brush was polymerized in each layer.

The static water contact angles of poly (DEGMA_90_-co-OEGMA_10_) and poly (HEMA_20_-co-OEGMA_80_) were measured at 43.2 ± 1.5° and 76.1 ± 2.4°, respectively. As expected, the poly (HEMA_20_-co-OEGMA_80_) layer containing hydrophilic HEMA had greater wettability than the poly (DEGMA_90_-co-OEGMA_10_). 

The weights of MLTRPB-Ti according to the number of layers are summarized in Table 1. The mean weight of 20 bare-titanium implants was 14.8293 g (average weight; 0.7415 g each) and for the four-layered MLTRPB (4-MLTRPB) was 2052.0 (±66.7) μg/cm^3^. With an increase in the number of polymer layers on the Ti implant, the thickness of the polymer brush increased to 3.075 (±0.134) μm. This resulted in increased loading of antibiotics, which was up to 37.3 μg/cm^3^ GM and 17.9 μg/cm^3^ LVF in 4-MLTRPB. Presumably due to the use of a HEMA-rich polymer brush, the GM resulted in a greater amount loaded in the MLTRPB than in the LVF owing to its hydrophilic nature. FE-SEM images of the MLTRPB-Ti implants with different numbers of layers are shown in Figure 4.

### 3.2. Drug Release Properties

The increased release behavior of antibiotics was observed as the thickness of MLTRPB-Ti increased. The released doses of LVF and GM from different layers of MLTRPB-Ti at 25 °C and at 40 °C are shown in Figure 5. In the first 6 h, LVF released up to 96.7% (17.03 μg cm^−^^2^) of the loaded amount at 40 °C, whereas it was 56% (9.13 μg cm^−^^2^) when below LCST (25 °C). The maximum release amount of GM was 33.63 μg cm^−^^2^ (90.1%) at 40 °C, which decreased to 23.2% at a temperature below LCST (25 °C). These findings demonstrate the release behavior of the antibiotic owing to a change in the morphology of the temperature-responsive polymer.

### 3.3. In Vitro Antibacterial Effect

The MBC of LVF and GM against *S. aureus* was in 4 μg mL^−^^1^ and 2 μg mL^−^^1^, respectively, which was similar to previous reports [20,30]. The bare implant incubated with *S. aureus* at 37 °C showed biofilm formation on the implant surface after 24 h (Figure 6). When tested for antibacterial activity, one, two, and three layers of MLTRPB-Ti-LVF had no significant effect, presumably due to an insufficient amount of LVF loaded on MLTRPB-Ti, but 4-MLTRPB-Ti-LVF showed a significant antibacterial effect at 40 °C, a temperature above LCST (Figure 7). Compared to MLTRPB-Ti-LVF, MLTRPB-Ti-GM screws had a greater amount of antibiotics impregnated in the polymer brushes. (Figure 5C) Of note, the MBC of GM is lower than that of LVF and therefore larger dose of impregnated GM resulted in a greater antibacterial effect of MLTRPB-Ti-GM. Below LCST (25 °C), both LVF- and GM-loaded MLTRPB screws had no antibacterial activity. The IVIS results also showed a decreased luminescence signal in correlation with the number of coated layers. 

In summary, LVF had a significant antibacterial effect on 4-MLTRPB-Ti, while both 3- and 4-MLTRPB-Ti had an effect when combined with GM. This result suggests that at least three-layered MLTRPB is necessary to have a sufficient antibacterial effect, but fewer layers can also yield successful outcomes if used in conjunction with hydrophilic antibiotics.

### 3.4. Cytotoxicity of MLTRPB

MLTRPB showed no inhibition of cell growth even at high concentrations of poly (DEGMA_90_-co-OEGMA_10_), whereas poly(HEMA_20_-co-OEGMA_80_) at high concentrations (500 ng/μL) showed 91.4% cell viability. However, at lower concentrations, no cytotoxicity was observed, indicating that MLTRPB is a biocompatible polymer and is not toxic to osteoblast cells (Figure 8).

### 3.5. Effect of MLTRPB In Vivo

Of the 40 animals that underwent surgery, one rat in group III died during the surgical intervention, leaving a total of 39 rats for analysis. Following euthanasia fourteen days after the surgery, evident infective tissue was observed when the skin fold was removed in all rats (Figure 9A–C). When the histology of pericapsular soft tissue samples was analyzed, abundant inflammatory cells were observed in all samples, which indicates that there was a limited antibacterial effect from eluted antibiotics from the MLTRPB coating on the pericapsular tissue (Figure 9D–F). 

CFUs on MLTRPB-Ti showed a greater reduction in bacterial bioburden in correlation with the number of MLTRPB layers applied. As compared to a non-coated implant, 1-MLTRPB resulted in a 77.2% reduction in CFUs, while 3-MLTRPB resulted in 99.8% and 4-MLTRPB resulted in more than 99.9% of CFUs. (Figure 10). Both 3-MLTRPB and 4-MLTRPB resulted in a nearly three-log reduction in bacterial biofilm, which is considered clinically significant. It should be noted that two out of nine in group 3 and seven out of 10 in group 4 showed no evidence of any bacterial biofilm on the implant.

In the current study, temperature-responsive polymer brushes with LCST in the range of 38–40 °C were polymerized and attached to the Ti implant surface. The MLTRPB-Ti implant was then impregnated with antibiotics, and the antibacterial properties of this strategy were investigated. The result was promising, as MLTRPB substantially decreased the bacterial bioburden on the implant, which correlated with the number of layers. Moreover, the use of a four-layer MLTRPB resulted in a three-log reduction in bacterial biofilms, which is considered to be significant in clinical practice [31]. 

Polymer brushes utilized in the current coating method are known to be nontoxic and have been extensively studied to use as a biocompatible targeted drug delivery system [32,33]. To design this novel thermo-responsive coating, we focused on multi-layered films comprising linear polymer “brush” layers, which were fabricated with two-, three-, and four-layer brush architectures. This strategy was suggested from previous literature, which validated the amplification of surface-grafted polymer films in aqueous media by covalently layering thermosensitive brushes [34,35]. In order to function as a scaffold for bio-application, the structure of a multi-layered polymer was designed and fabricated to respond to external stimuli, in our case temperature. Unlike physical bonding, chemically bonded layer-by-layer structures can avoid structural instability [36]. Our MLTRBPs are temperature responsive and collapse above the LCST, thus releasing impregnated antibiotics when the local temperature increases [37]. In addition, the two polymer brushes used in fabricating the MLTRPB, poly (DEGMA_90_-co-OEGMA_10_) and poly (HEMA_20_-co-OEGMA_80_), have different polarities, resulting in varying antibacterial effects according to the hydrophilicity of the antibiotics. Our MLTRPB was more effective when used in combination with water-soluble antibiotics, in our case GM. This characteristic may suggest that MLTRPB can be applied in other antibiotics, especially with a hydrophilicity nature. For example, fluconazole, an antifungal agent used widely to treat fungus infection may also be used with this coating considering its hydrophilicity. However, such a hypothesis would require further investigation for validation.

While the current study demonstrated the in vitro and in vivo efficacy of MLTRPB, it also suggests the potential limitation of this coating method in its current form. The antibiotics impregnated in the MLTRPB seem to provide a sufficient dose to prevent bacteria from adhering to the implant, but this dose seems insufficient to eradicate the infection in the surrounding soft tissue. However, this finding was anticipated, as biofilm can be formed not only in the implant but also in the fibrotic tissue or on the bony structures. Therefore, once the entire antibiotic is released, it is likely that the biofilm on the soft tissue eventually leads to periprosthetic infection. We believe that the addition of more layers is an effective solution. However, as the main goal of inserting the implant in orthopedic surgery is to achieve bone healing, the effect of a multicoating layer on osteointegration needs to be further validated. In addition, there are numerous factors that may contribute to the release behavior of the thermos-responsive coating other than the fever from infection. Inflammatory response from the surgery or peri-implant diseases initiated from metal debris may also initiate a temperature rise, and various atmospheres, such as acidity, may influence hydrophilicity and thereby interfere with the drug loading and release but such conditions are not taken into account in the current study [38,39,40]. Lastly. The current coating method needs further validation to test its interference with bone healing and osteointegration which is an important prerequisite of the implant when applied in orthopedics surgery.

Nonetheless, our novel coating method showed a highly effective antibacterial effect with biocompatibility and selectivity of action at physiological temperatures, a trait that is necessary for smart antibacterial surfaces for in vivo application [41,42]. Additionally, the multi-layered coating increased the antibiotics loading dose thereby demonstrating a greater antibacterial effect. The method provided the potential to be used in combination with various antibiotics, especially with high hydrophilicity, which can be applied to protect infections from various bacterial and fungus strains.

## 4. Conclusions

The current study demonstrated that MLTRPB based on poly (DEGMA_90_-co-OEGMA_10_) and poly (HEMA_20_-co-OEGMA_80_) may provide promising antibiotic impregnation and release properties. Owing to the hydrophilic HEMA-rich layer, the novel MLTRPB was more effective when used in conjunction with hydrophilic antibiotics (GM). This strategy has resulted in the successful eradication of adherent bacterial biofilms in vivo.

## Data Availability

The dataset used and analyzed during the current study are available from the corresponding author on a reasonable request.

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
