# Peer review of "Application of Multi-Layered Temperature-Responsive Polymer Brushes Coating on Titanium Surface to Inhibit Biofilm Associated Infection in Orthopedic Surgery"

_polymers, 2022, doi:10.3390/polym15010163_

Round 1

Reviewer 1 Report

This is an interesting study about temperature-responsive polymer brush coating on titanium surfaces to inhibit biofilm-associated infection in orthopedic surgery. I suggest it for publication after the following points are addressed.

1. The text in scheme 1, figures 1, 2, 3, 5, and 6 should be larger and clearer.

2. The quality of figure 10 should be improved.

3. The affiliations of the authors should be completed.

4. It is better to include the chemical structures of antibiotics in scheme 1.

5. In the introduction, other approaches to bacteria inhibition should be added. Several studies (Acta Biomaterialia 7 (5), 2053-2059; Langmuir  2019, 35, 52, 17027-17036) are recommended to be included.

6. Formatting issues. Line 204, 'Measuring Bacterial bioburden' to 'Measuring Bacterial Bioburden'; line 242, 'Table A90. co-OEGMA10)'. Please check all. 

Author Response

We would like to first thank the reviewer for the important comments and reviews. We were able to strengthen our paper based on the reviewer’s important comments.

Our responses to the reviewer's comments and reviews are as follows:

1. The text in scheme 1, figures 1, 2, 3, 5, and 6 should be larger and clearer.

-> We have added Scheme 1 and figure 1,2,3,5,6 as separate files to increase clarity.

2. The quality of figure 10 should be improved.

-> Figure 10 has been modified so that it better elucidates the result of the in vivo study.

3. The affiliations of the authors should be completed.

-> We have completed the affiliation of the authors.

4. It is better to include the chemical structures of antibiotics in scheme 1.

-> Thank you for the important comment. We also think including the chemical structure of the antibiotics (levofloxacin and gentamicin) will help the readers understand the manuscript. We have added this in scheme 1-C.

5. In the introduction, other approaches to bacteria inhibition should be added. Several studies (Acta Biomaterialia 7 (5), 2053-2059; Langmuir  2019, 35, 52, 17027-17036) are recommended to be included.

-> We thank the reviewer for this important comment and for suggesting relevant papers. We agree that the approaches to bacteria inhibition should be more thoroughly discussed. Following the reviewer’s recommendation, we added other approaches that have been utilized to inhibit bacteria and included the methods suggested in Acta Biomaterialia and Langmuir.

6. Formatting issues. Line 204, 'Measuring Bacterial bioburden' to 'Measuring Bacterial Bioburden'; line 242, 'Table A90. co-OEGMA10)'. Please check all.

-> Thank you for your comment. We corrected the formatting issue and also checked carefully for any other flaws throughout the manuscript.

Additional changes have been made to upgrade the introduction and discussion section, which is in red letters. 

Reviewer 2 Report

Comments on Choi et al:

The aim of this manuscript is to use a strategy of fabricating polymer brushesin multilayer architectures featuring diversehydrophilicities and compositionsto optimize antibiotic delivery to implant surfaces.

This manuscript shows rich content, providing a deep insight for some works: the study is within the journal’s scope, and I found it to be well-written, providing sufficient information. Even if the manuscript provides an organic overview, with a densely organized structure and based on well-synthetized evidence, there are some suggestions necessary to make the article complete and fully readable. For these reasons, the manuscript requires major changes.

Please find below an enumerated list of comments on my review of the manuscript:

Line 194: the sentence “which was divided into five groups of 10” should be corrected in “which was divided into four groups of 10”

The antibiotic release is temperature responsive, so not specific and not linked to the biofilm; there should be consequently some cases in which the antibiotic is released without the presence of bacteria.

Some kinds of inflammatory processes not biofilm-linked should be mentioned; an example is the peri-implant disease, due to metal particles deposition from surgical bars, as suggested by recent evidence (see, for reference: Falisi, G.; Foffo, G.; Severino, M.; Di Paolo, C.; Bianchi, S.; Bernardi, S.; Pietropaoli, D.; Rastelli, S.; Gatto, R.; Botticelli, G. SEM-EDX Analysis of Metal Particles Deposition from Surgical Burs after Implant Guided Surgery Procedures. Coatings 202212, 240. https://doi.org/10.3390/coatings12020240). 

The main topic is interesting, and certainly of great clinical impact. As regards the originality and strengths of this manuscript, this is a significant contribute to the ongoing research on this topic, as it extends the research field on the application of polymer brushes in multilayer architectures featuring diverse hydrophilicities and compositions to optimize antibiotic delivery to implant surfaces. Overall, the contents are rich, and the authors also give their deep insight for some works.

As regards the section of methods, there is a specific and detailed explanation for the methods used in this study: this is particularly significant, since the manuscript relies on a multitude of methodological and statistical analysis, to derive its conclusions. The methodology applied is overall correct, the results are reliable and adequately discussed.

The conclusion of this manuscript is perfectly in line with the main purpose of the paper: the authors have designed and conducted the study properly. As regards the conclusions, they are well written and present an adequate balance between the description of previous findings and the results presented by the authors.

Finally, this manuscript also shows a basic structure, properly divided and looks like very informative on this topic. Furthermore, figures and tables are complete, organized in an organic manner and easy to read.

In conclusion, this manuscript is densely presented and well organized, based on well-synthetized evidence. The authors were lucid in their style of writing, making it easy to read and understand the message, portrayed in the manuscript. Besides, the methodology design was appropriately implemented within the study. However, many of the topics are very concisely covered. This manuscript provided a comprehensive analysis of current knowledge in this field. Moreover, this research has futuristic importance and could be potential for future research. However, major concerns of this manuscript are with the introductive and discussion sections: for these reasons, I have major comments for these sections, for improvement before acceptance for publication. The article is accurate and provides relevant information on the topic and I have some major points to make, that may help to improve the quality of the current manuscript and maximize its scientific impact. I would accept this manuscript if the comments are addressed properly.

Author Response

We would like to first thank the reviewer for the important comments and reviews. We were able to strengthen our paper based on the reviewer’s important comments.

Our responses to the reviewer's comments and reviews are as follows:

Please find below an enumerated list of comments on my review of the manuscript:

Line 194: the sentence “which was divided into five groups of 10” should be corrected in “which was divided into four groups of 10”

->  Thank you for the comment. We have wrongfully described this, and this has been corrected.

The antibiotic release is temperature responsive, so not specific and not linked to the biofilm; there should be consequently some cases in which the antibiotic is released without the presence of bacteria. Some kinds of inflammatory processes not biofilm-linked should be mentioned; an example is the peri-implant disease, due to metal particles deposition from surgical bars, as suggested by recent evidence

-> Thank you very much for the important comment. This is an important point we have missed and should have been considered. We will implement this comment in our future study. For now, we have included this as a limitation of the study in the discussion section.  

However, major concerns of this manuscript are with the introductive and discussion sections: for these reasons, I have major comments for these sections, for improvement before acceptance for publication

-> We would like to thank the reviewer for the important and thorough review. Following the reviewer’s comment, we have implemented the reviewer's comment and added new content and references to the introduction and the discussion section.

Changes made to upgrade the introduction and discussion section are in red letters. 

Reviewer 3 Report

The rationale for this work and the design approach seems reasonable. Nevertheless, the manuscript also has several methodological issues and few minor editorial mistakes as highlighted below:

1. Why only S. aureus strain was chosen to study the antimicrobial activity? Any other strains studied for the effect of the coating?

2.  As described in section 2.2: The loading process was performed under vacuum for 1 h, followed by drying under a nitrogen stream. Is the vaccum responsible for loading of drug into the implant? Describe in detail mechanism of drug loading.

3. What set-up was used for drug release from the implant?

4. What is the analytical range, LoD, LoQ, and specificity of the developed UV method? Non-separation methods such as UV have several drawbacks for application in bioanalysis. Please justify the use of UV method in this work.

5. Please highlight the novelty of this work. How is the work an incremental advancement of reported coated implant? 

6. Can authors highlight on the benefit of using the developed coated implant vs. any other form to avoid antifungal resistance? This can also be a part of the introduction.

Author Response

We would like to first thank the reviewer for the important comments and reviews. We were able to strengthen our paper based on the reviewer’s important comments.

Our responses to the reviewer's comments and reviews are as follows:

1. Why only S. aureus strain was chosen to study the antimicrobial activity? Any other strains studied for the effect of the coating?

-> Thank you for the important comment. The majority of infection in the orthopedic field is caused by S aureus, and the bacterial strain in the current study is commonly used to test periprosthetic joint infection both in vitro and in vivo. The strain has also been validated to reproduce biofilm consistently (Carli et al J Bone Joint Surg Am 2016 Vol. 98 Issue 19 Pages 1666-1676). We have included this in the manuscript with the relevant references. Due to our success with S aureus, we are currently undergoing similar studies using different bacterial strains (gram negative E coli) and would like to report our results in the near future.

2. As described in section 2.2: The loading process was performed under vacuum for 1 h, followed by drying under a nitrogen stream. Is the vaccum responsible for loading of drug into the implant? Describe in detail the mechanism of drug loading.

-> The swelling of the MLTRPB occurs, up to 160-300% based on our previous study, when encountered with the water molecule. This results in the release of antibiotics. Therefore, complete drying will allow for maximizing drug loading into the coating. As the hydrogen bonding between antibiotics and polymer brush influences the antibiotic loading dose, the vacuum drying process allows complete dehydration of MLTRPB which provides a uniform loading atmosphere for each sample. We have implemented this in the manuscript.

3. What set-up was used for drug release from the implant?

-> The more detailed setup protocol is now included in 2.2 which is described as follows: The LCST was confirmed by testing antibiotic release from MLTRPB-Ti implants at different temperatures (25, 40℃). MLTRPB-Ti with LVF was loaded into the 5ml of pH 7.0 PBS buffer, to imitate the inflammatory atmosphere. 0.5ml of the suspension was retrieved at 0.5, 1, 2, 4, and 6 hours and the concentration of LVF release (μg mL-1) was measured using fluorescence spectroscopy (Cary Eclipse spectrometer, Agilent, USA) at excitation and emission wavelengths of 292 nm and 540 nm and calculated using an established calibration curve [21]. The GM released from MLTRPB-Ti into PBS buffer-d2 (PBS was dissolved in D2O) at pH 7.0 was estimated at 0.5, 1, 2, 4, and 6 hours using FT-NMR (JEOL 300 MHz, Japan) by measuring the peak area (1.84 ppm), which belongs to GM, at each time points.

4. What is the analytical range, LoD, LoQ, and specificity of the developed UV method? Non-separation methods such as UV have several drawbacks for application in bioanalysis. Please justify the use of UV method in this work.

-> We thank the reviewer for the important comment. We agree that there is several drawbacks to using UV for bio-analysis. However, we would like to point out that there is only a limited method for detecting levofloxacin, and use of spectroscopy is widely utilized for quantification. As the method was used in a number of previous literature, we followed their protocol to quantify levofloxacin in our study. We were unable to find the data on LoD or LoQ for using spectroscopy for the quantification of LVF, but based on the study by Dafale et al, (Journal of Pharmaceutical Analysis, Volume 5, Issue 1, 2015, 18-26), the test has the accuracy of 101.23% with relative standard deviation (RSD) of 0.72%, which confirms the ability of the method to determine the Levofloxacin concentration within the range of 80%−120%. The other methods suggested to quantify LVF, such as HPLC, do not necessarily provide more accurate results. Once again, this is an important comment, but we ask for the reviewer’s understanding that this is a widely used method to detect LVF and is reproducible. We have included the explanation in the material & method section.

5. Please highlight the novelty of this work. How is the work an incremental advancement of reported coated implant? 

-> Thank you for the important comment.  We have added how current work may incremental advancement of reported coated implant in the discussion section.

6. Can authors highlight on the benefit of using the developed coated implant vs. any other form to avoid antifungal resistance? This can also be a part of the introduction.

-> Thank you for the important comment. Fungus is one important source of infection following orthopedic surgery, but currently, fungi constitute only about 1% of infections. Therefore, the current study focused on a more common bacterial strain, S aureus. Theoretically, the high dose of antifungal agent delivered to the infection site may also eradicate or at least decrease the bioburden, but the authors have limited knowledge of antifungal agents, and even with the literature review, we were unable to provide sufficient evidence to answer to the reviewer’s comment/question. However, we found one of the antifungal agents, amphotericin, is hydrophilic in nature with a molecular mass similar to LVF. Therefore, the application of this antifungal agent to our coating seems promising. However, this is only a hypothesis and needs a further valuation. We explained this in the discussion section.

Additional changes have been made to upgrade the introduction and discussion section which is in red letters. 

Round 2

Reviewer 2 Report

Dear Authors, even though I have found You tried to address to all suggestions, still I miss the point which should be discussed on the periimplant diseases not biofilm related with the suggested and recent evidences.

Please complete the revisions. 

Regards 

Author Response

We apologize for not making a comprehensive response in the previous round. We have added relevant references and content according to the reviewer's comment. We thank the reviewer for the comment. 

Reviewer 3 Report

The authors have addressed the concerns raised during the review process. Now, in the current form, the manuscript can be accepted for publication

Author Response

We thank the reviewer for the review.